# Impact of Permafrost Thaw and Climate Warming on Riverine Export Fluxes of Carbon, Nutrients and Metals in Western Siberia

**Oleg S. Pokrovsky** [1,2,3,*], **Rinat M. Manasypov** [1] , **Sergey G. Kopysov** [4], **Ivan V. Krickov** [1] , **Liudmila S. Shirokova** [2,3], **Sergey V. Loiko** [1] , **Artem G. Lim** [1] , **Larisa G. Kolesnichenko** [1], **Sergey N. Vorobyev** [1] **and Sergey N. Kirpotin** [1]

1   BIO-GEO-CLIM Laboratory, Tomsk State University, 36 Lenina ave, 634050 Tomsk, Russia;
    rmmanassypov@gmail.com (R.M.M.); krickov_ivan@mail.ru (I.V.K.); s.loyko@yandex.ru (S.V.L.);
    lim_artyom@mail.ru (A.G.L.); klg77777@mail.ru (L.G.K.); soil@green.tsu.ru (S.N.V.);
    kirp@mail.tsu.ru (S.N.K.)
2   Geosciences and Environment Toulouse, UMR 5563 CNRS, 14 Avenue Edouard Belin,
    31400 Toulouse, France; liudmila.shirokova@get.omp.eu
3   N. Laverov Federal Center for Integrated Arctic Research, Russian Academy of Sciences,
    23 Naberezhnaya Sev. Dviny, 163000 Arkhangelsk, Russia
4   Institute of Monitoring of Climatic and Ecological Systems SB RAS, 10/3 Academichesky ave,
    634055 Tomsk, Russia; wosypok@mail.ru
*   Correspondence: oleg.pokrovsky@get.omp.eu

**Abstract:** The assessment of riverine fluxes of carbon, nutrients, and metals in surface waters of permafrost-affected regions is crucially important for constraining adequate models of ecosystem functioning under various climate change scenarios. In this regard, the largest permafrost peatland territory on the Earth, the Western Siberian Lowland (WSL) presents a unique opportunity of studying possible future changes in biogeochemical cycles because it lies within a south–north gradient of climate, vegetation, and permafrost that ranges from the permafrost-free boreal to the Arctic tundra with continuous permafrost at otherwise similar relief and bedrocks. By applying a "substituting space for time" scenario, the WSL south-north gradient may serve as a model for future changes due to permafrost boundary shift and climate warming. Here we measured export fluxes (yields) of dissolved organic carbon (DOC), major cations, macro- and micro- nutrients, and trace elements in 32 rivers, draining the WSL across a latitudinal transect from the permafrost-free to the continuous permafrost zone. We aimed at quantifying the impact of climate warming (water temperature rise and permafrost boundary shift) on DOC, nutrient and metal in rivers using a "substituting space for time" approach. We demonstrate that, contrary to common expectations, the climate warming and permafrost thaw in the WSL will likely decrease the riverine export of organic C and many elements. Based on the latitudinal pattern of riverine export, in the case of a northward shift in the permafrost zones, the DOC, P, N, Si, Fe, divalent heavy metals, trivalent and tetravalent hydrolysates are likely to decrease the yields by a factor of 2–5. The DIC, Ca, $SO_4$, Sr, Ba, Mo, and U are likely to increase their yields by a factor of 2–3. Moreover, B, Li, K, Rb, Cs, $N-NO_3$, Mg, Zn, As, Sb, Rb, and Cs may be weakly affected by the permafrost boundary migration (change of yield by a factor of 1.5 to 2.0). We conclude that modeling of C and element cycle in the Arctic and subarctic should be region-specific and that neglecting huge areas of permafrost peatlands might produce sizeable bias in our predictions of climate change impact.

**Keywords:** river flux; weathering; organic matter; permafrost; trace element; river

## 1. Introduction

Arctic warming is anticipated to result in massive carbon (C) mobilization from permafrost peat to atmosphere, rivers and lakes, thereby potentially worsening global warming via greenhouse gases (GHG) emissions [1,2]. Permafrost peatlands cover roughly 2.8 million km$^2$ or 14% of permafrost-affected areas, mostly in Northern Eurasia (Bolshezemelskaya Tundra, $0.25 \times 10^6$ km$^2$; western Siberia, $1.05 \times 10^6$ km$^2$; Northern Siberia and Eastern Siberia lowlands, $0.5 \times 10^6$ km$^2$) and contain a huge amount of highly vulnerable carbon in soil and surface waters [3]. Except for several regional studies of peatland lakes and small streams in Canada [4–6] and northern Sweden [7], the control factors, timing and reality of C and related elements release from soil and sediments are largely unknown, in part because element export by rivers across the permafrost peatlands is still poorly quantified.

One of the largest peatlands in the world is the Western Siberia Lowland (WSL) which exhibits a disproportionally high contribution to C storage and exchange fluxes [8] and presents a prominent exception to well-studied aquatic and soil ecosystems in mountainous territories of Northern America and Scandinavia. Specific features of the WSL are: (1) developed on flat, low runoff terrain with long water residence time in both lentic and lotic systems; (2) acts as important C stock in the form of organic-rich histosols (peat soils); (3) emits substantial amount of $CO_2$ and $CH_4$ to the atmosphere from the surface of inland waters; and (4) exhibits high dissolved organic carbon (DOC) concentration and low pH in waters in contact with peat soils. These factors determine rather unique and still poorly known aquatic communities in dystrophic to mesotrophic peatland waters. Due to the lack of nutrients, shallow photic layer and high sensitivity to water warming, aquatic ecosystems of frozen peatlands are highly vulnerable to the permafrost thaw and can respond unexpectedly to ongoing climate warming in terms in their C, nutrient, and metal storage and export fluxes as well biodiversity and organisms adaptation strategies.

Western Siberia is a key region for biogeochemical studies in the Arctic [2] as this region includes the continuous-discontinuous permafrost transition and experiences substantial thermokarst. As a result, this high-priority region is most susceptible to thaw-induced change in solute transport by rivers and its export to the ocean and atmosphere, notably via activation of deep underground flow [9] and exposure of large volumes of previously frozen peat and mineral soils [10–12]. Raised by incontestable Arctic amplification of overall climate change, the fate of carbon, nutrient and metal in Arctic rivers is at the forefront of field and modeling studies [13–20]. In particular, the Arctic Great Rivers Observatory (GRO) program assessed concentrations and export fluxes of 6 largest Arctic Rivers including the Ob River, draining sizeable part of the WSL [21,22]. However, widespread loss of hydrological monitoring in the beginning of 21st century was especially pronounced for small and medium-size rivers of the permafrost-affected part of the WSL [23]. At the same time, the Ob River is not suitable for modeling of possible changes in western Siberia because (1) it is strongly influenced by its largest permafrost-free tributary, the Irtysh River; and (2) it cannot be used for assessing the fluxes of northern (permafrost) zones as it integrates vast territory of variable permafrost coverage (20% in average) and landscape parameters. For these reasons, the study of small rivers at the WSL territory is more suitable for assessing both mechanisms of flux formation and its possible future changes.

Currently, a dominant paradigm is that riverine fluxes of C and inorganic nutrients are increasing in virtually all permafrost-affected rivers [14,24,25]. In particular, the increase in suspended versus dissolved transport of elements may be due to abrasion of lake shores and riverbanks. At the same time, enhanced groundwater input will lead to an increase in the transport of truly dissolved forms. Recently, following the pioneering work of Frey et al. [26–28], the concentrations of dissolved, particulate and colloidal carbon, nutrient and metals in WSL rivers, lakes and soil waters have been studied over a sizable latitudinal gradient [29–38]. These results allowed a first-order assessment of the consequences of climate warming and permafrost thaw on river water concentrations of dissolved and particulate forms of elements from western Siberia to the Arctic Ocean and C emission to the atmosphere. For this, a "substituting space for time" approach was employed, which postulates, in a broad context, that spatial phenomena which are observed today can be used to describe past and future events [39].

Thus, a concentration pattern of major and trace elements (TE) in WSL river suspended matter implies that, upon a progressive shift of the permafrost boundary northward, there will be a sizeable decrease in concentrations of alkalis, alkaline-earths, divalent heavy metals, and trivalent and tetravalent hydrolysates in northern rivers in currently discontinuous and continuous permafrost zones [31]. This decrease may be by a factor of 2–5 from the position of the minimum elemental concentration in sporadic to discontinuous permafrost zones relative to the continuous permafrost zone. Concerning the dissolved element concentration and potential transport in the WSL rivers, the following results were achieved implying the substituting space for time scenario. From the permafrost thaw perspective, the increase in depth of the active layer and connectivity of a river with underground water reservoirs may decrease the colloidal fraction (1 kDa–0.45 µm) of OC, Fe, Al and number of divalent metals as well as low-soluble trivalent and tetravalent hydrolysates in the sporadic and isolated permafrost zone as it becomes permafrost-free [30]. The forestation of wetlands and lake drainage may slightly diminish colloidal transport of DOC and metals at the expense of low molecular weight forms. Overall, given the significant role of seasonal and forestation effects on colloidal forms of OC and TE in WSL rivers, major changes in the speciation of riverine C and nutrients in the WSL may occur due to changes in vegetation rather than in temperature and precipitation [30]. A northward permafrost boundary shift with increase in air and water temperature may decrease or maintain, rather than increase of major nutrient (K, P, N, Si) and DOC concentrations in rivers draining through continuous permafrost zone [37].

All these predictions described above were made based solely on evolution of concentrations of soluble and particulate C, metals and nutrients in WSL rivers, without taking into account the hydrological flux. The latter could not be addressed until now due to lack of reliable information on river discharge over different seasons. The present work aims to fill this gap by quantifying the export fluxes of ~30 WSL rivers of various size, combined with new hydrological modeling of region and season-specific river runoff of the WSL territory. The chosen rivers encompass a large gradient of climate, vegetation and permafrost distribution and thus enables a substituting space for time scenario to provide a tentative prediction of possible changes in riverine fluxes for short- and long-term prospectives. The present work is built on seasonally resolved sampling performed in 2015 and 2016, and incorporates thorough hydrological modeling to calculate the seasonal discharge. Building upon our previous studies of the WSL rivers [29–31,33,34,37], here we assess for the first time, element export fluxes (yields) across a large permafrost-affected territory and large number of rivers. The obtained elementary yields are essential for further modeling of biogeochemical cycles in the permafrost regions.

## 2. Study Site and Methods

### 2.1. Rivers of Western Siberian Lowland, their Sampling and Analyses

The 32 rivers of the Ob, Pur and Taz watersheds in the WSL were sampled (Figure 1). Detailed climatic, lithological and physio-geographic characteristics for the WSL are presented elsewhere [29,33,34]. The dominant lithology is Quaternary deposits (silt, clay, sand) overlaid by peat. The climate gradient of sampled rivers presents a decrease in mean annual air temperature (MAAT) from −0.5 °C in the south (Tomsk region) to −9.5 °C in the north (Arctic coast). Annual precipitation is fairly constant ranging between 550 mm in the south and 600 mm at the lower reaches of the Taz River. The river runoff ranges from $190 \pm 30$ mm y$^{-1}$ in the south to $300 \pm 20$ mm y$^{-1}$ in the north [40]. The distribution of the permafrost reflects the south-north MAAT gradient and changes from isolated and sporadic in the south to discontinuous and continuous in the north. In 2015, we sampled rivers in spring (18 May–25 June) and summer (25 July–21 August). In 2016, sampling was performed in spring (17 May–15 June 2016), summer (1–29 August 2016), and autumn (24 September–13 October 2016). We followed the change of seasons during our sampling campaign and moved from the south to the north in spring and from the north to the south in autumn thus collecting the river water at approximately the same period of the discharge. Note that more frequent sampling would be desirable to accurately evaluate the annual export flux, but rather harsh environment and logistical issues

constrained sampling. In contrast, route sampling is a common way to assess chemical weathering in extreme environments [41,42], and it is accepted that single sampling during high flow season provides the best agreement with time-series estimates [43].

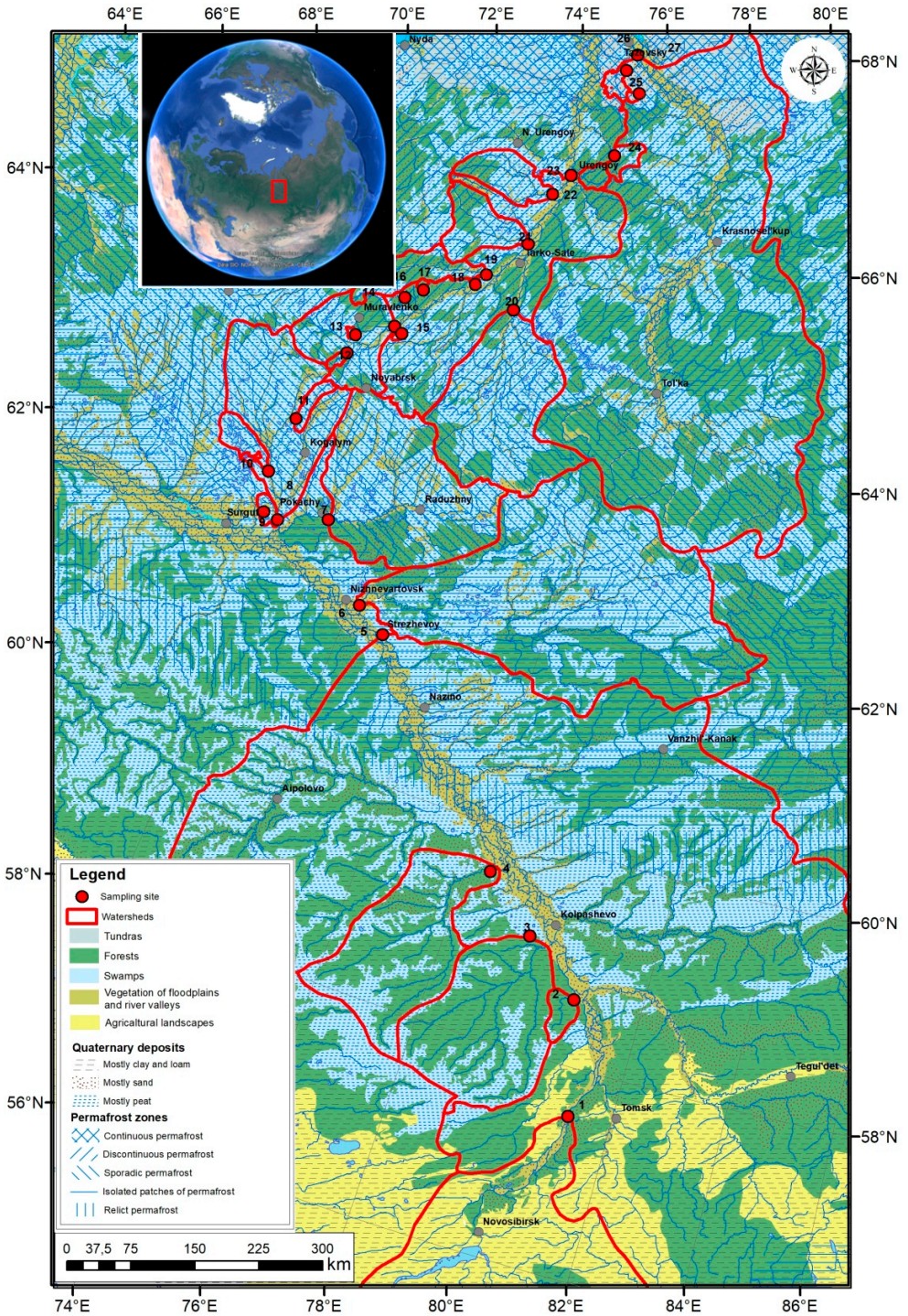

**Figure 1.** The WSL river sampling points. The numbers on the map correspond to the following rivers: (1) Ob' (Pobeda), (2) Maliy Tatosh, (3) Chaya, (4) Vyalovka, (5) Ob' (Aleksandrovskoye), (6) Vakh, (7) Agan, (8) Tromyegan. (9) Vach-Yagun, (10) Vachinguriyagun, (11) Pintyr'yagun, (12) Kamgayakha, (13) Khatytayakha, (14) Pyakupur, (15) Lymbyd'yakha, (16) Apoku-Yakha, (17) Etu-Yakha, (18) Seryareyakha, (19) Purpe, (20) Aivasedapur, (21) Tydylyakha, (22) Yamsovey, (23) Pur, (24) Ngarka Khadyta-Yakha, (25) M. Kheyakha, (26) Nuny-Yakha, (27) Taz. The upper left insert is from Google Maps®.

The two sampled years contrasted in mean monthly and annual temperature and precipitation. The summer 2015 was warm (+0.3 °C above normal for July), while summer 2016 was cool (−3.1° below normal for July). The summer 2015 was wet (180% of normal precipitation for July), and the summer 2016 was less wet (but still 130% of normal precipitation for July). The winter precipitations of 2014–2015 and 2015–2016 were rather similar and sizably higher (ca. 140%) than the normal winter values. The normal values here are defined as between 1970 to 2000 based on the Roshydromet archives [44].

The surface water was collected in a polypropylene 1-L container. Samples were filtered through 0.45 μm cellulose acetate filters using a Nalgene 250 mL filter unit combined with a vacuum pump. All filtrations were run on site, in a protected environment, within 2 h of river water collection. Immediately after filtration, samples for DOC, DIC, major and trace elements were stored in the refrigerator during 1–2 months prior to the analyses, while the samples for nutrients were kept frozen. Dissolved (<0.45 μm) concentrations of nutrients, major and trace elements in WSL river waters were analyzed as described elsewhere [33,34,37,45]. Major anion concentrations ($Cl^-$ and $SO_4^{2-}$) were measured by ion chromatography (HPLC, Dionex ICS 2000) with an uncertainty of 2%. DOC and DIC were analyzed using a Shimadzu TOC 6000 with an uncertainty of 3–5% [46]. For all major and most trace elements, analyzed by ICP MS, the concentrations in the blanks were below analytical detection limits (≤0.1–1 ng/L for Cd, Ba, Y, Zr, Nb, rare earth elements (REEs), Hf, Pb, Th, U; 1 ng/L for Ga, Ge, Rb, Sr, Sb; ~10 ng/L for Ti, V, total P ($P_{tot}$), Cr, Mn, Fe, Co, Ni, Cu, Zn, As). The international certified reference material SLRS-5 (Riverine Water Reference Material for Trace Metals) was used to validate the analysis. Further details of analytical uncertainties and detection limits for TE are provided elsewhere [30,34,45].

### 2.2. Hydrological Parameters, River Discharge Modeling, and Element Export Flux Calculations

The runoff was longitudinally modelled using average altitude of the watershed for gauged rivers [47]. For small and medium rivers draining palsa and polygonal bogs of the permafrost zone, we used empirical equations accounting for hydrological characteristics of these watersheds [48]. In this work, only open water period (May to October) was considered. The contribution of winter time export in large rivers typically does not exceed 10% of mean annual runoff [49]; moreover, small (<1000–10,000 km$^2$ watershed) WSL rivers freeze solid in winter [33]. We interpolated discharge by using watershed area change along the river mainstem. The river hydrographs were modelled using HBV Light program package (https://www.geo.uzh.ch/en/units/h2k/Services/HBV-Model.html) as described in refs [50–52]. For the gauged rivers, we modelled daily runoff based using mean water temperature and precipitation from adjacent meteostations. The data for reproducing the water discharge at key sites were taken from hydrological yearbooks and automatic information system of State monitoring of water bodies [53]. The daily air temperature and precipitation were taken from Russian Hydrometeocenter (URL: http://aisori.meteo.ru/ClimateR; [44]). The calibration of model parameters demonstrated that the HBV model adequately reproduces the seasonal dynamics of runoff (quality criteria of 0.75 to 0.90). The uncertainties of modelled discharge at ungauged rivers stem from several factors. First, there is uncertainties in characterizing watershed boundaries for the very flat, but poorly resolved, WSL territory. For small rivers, this uncertainty can amount to 30% due to lack of precise topographic information. For large rivers, it can be up to 7% due to difficulties in determination of exact position of the watershed divide on flat bog-lake landscapes. The second cause of uncertainty is a discrepancy between the model parameters of analogous rivers due to intrinsic differences in the conditions of runoff formation (up to 25%). Finally, the amount of precipitation obtained from the nearest meteostations might substantially differ from the actual precipitation at a given watershed, which is reflected, in particular, by low quality of the rain-flood modeling. The maximal overall uncertainty of daily discharges is determined by the most probable value of 30%. However, because we used two-month averaged discharges for calculating elementary yields, the real uncertainty of export fluxes is determined by the uncertainty of watershed delineations and ranges from 7 to 30%.

In order to calculate element export fluxes, we defined six latitudinal belts (56–58° N, 58–60° N, 60–62° N, 62–64° N, 64–66° N and 66–68° N). These latitude belts were selected based on: i) the permafrost map of the WSL, where the permafrost zones roughly follow latitude, and ii) necessity to integrate sufficient number of rivers in each permafrost zone and for statistical comparison. For statistical comparisons, we separated the years 2015 and 2016, because the autumn was sampled only in 2016. The number of rivers used for the latitudinal-average flux calculation ranged from 2 to 10 in 2016 and from 2 to 16 in 2015 for each latitudinal range.

## 3. Results

### 3.1. Impact of the Watershed Size and Season on Element Export Fluxes

The yields of DOC, representative major solutes (DIC, Ca, Mg) and nutrients (Si, $P_{tot}$, K, Fe) are illustrated in Supplementary Figure S1. Flux magnitudes exhibited sizable variation, ranging over one (DOC, Si and Mg) to two (DIC, Ca, K, Fe and $P_{tot}$) orders of magnitude. Generally, the variations were the highest for rivers having watersheds between 100 and 1000 km$^2$, for both permafrost-affected and permafrost-free. The river watershed area ($S_{watershed}$) exhibited rather weak control on elementary yields, as also illustrated by statistical treatment of individual seasons and full data set for both years (Supplementary Table S1). Overall, the watershed size exhibited more pronounced correlations with element yield in the permafrost-free zone compared to the permafrost zone. The correlations were quite low during spring flood but become pronounced during summer and autumn baseflow. The highest correlations were observed in summer, when Cl, SO$_4$, Fe, Cu, Y, Mo, Sb, REEs and Th were positively linked to $S_{watershed}$ ($R_{Spearman} > 0.58$, $p < 0.01$) in the permafrost-free zone.

Partial contribution of spring, summer, and autumn (2 months each) to the overall export of elements during the six-month open-water period (May to October) in 2016 demonstrated the dominant role of spring for DOC, Al and Fe (>60%), for both permafrost-free and permafrost-affected rivers (Figure 2). In contrast, soluble highly mobile elements (DIC, Na, K, Ca, Mg, Si) had less than 40% and 50% contribution of spring in the permafrost-free and permafrost zone, respectively, and a 2–3 times higher contribution during summer baseflow period compared to during the spring flood period. Nutrients (e.g., K, $P_{tot}$) and metals (e.g., Mn) presented an intermediate case with a half of yield occurring during the spring flood period and with negligible (<10%) role of autumn period in permafrost-free zone but comparable contribution of summer and autumn in the permafrost-affected part of the WSL.

Element export as a function of watershed area (Supplementary Figure S1) and season (Supplementary Figure S2) represent two main groups, namely (1) soluble highly mobile elements (Cl, SO$_4$, Sr, Rb, Ba and As, Sb, Mo, U (permafrost-free zone only) and (2) DOC and low soluble elements, typically present in the form of organic- and organo-mineral colloids especially in the permafrost zone [30], such as micronutrients (Fe, Co, Ni, Cu, Zn, V), toxic metals (Cr, Cd, Pb), Nb, and trivalent (Al, Ga, Y, REE) and tetravalent (Ti, Hf, Th) hydrolysates. Generally, the share of spring flood contribution in overall open-water export of elements was 20 ± 10% higher in permafrost-affected rivers than in rivers of the permafrost-free zone.

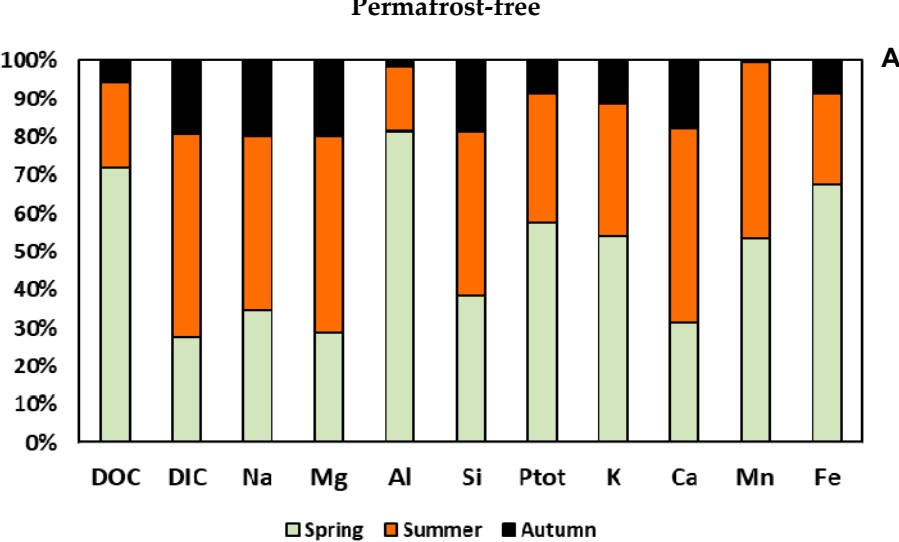

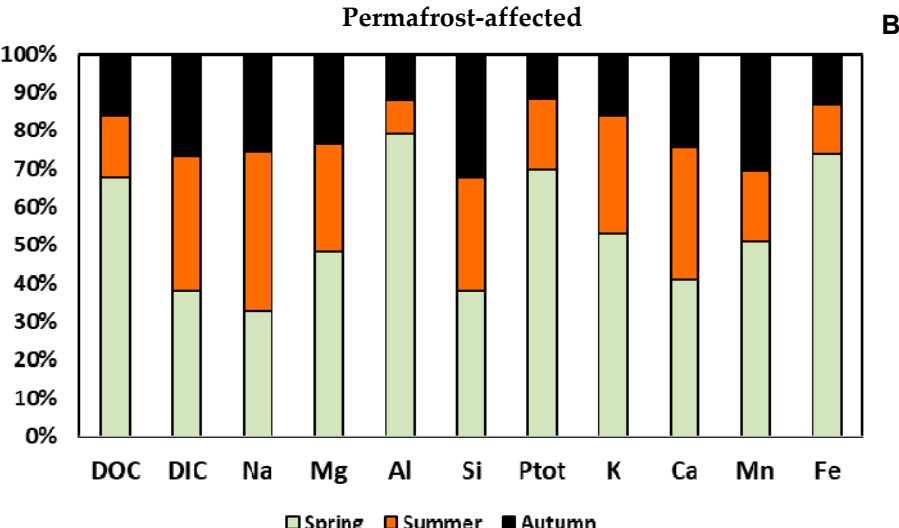

**Figure 2.** Partial contribution of spring, summer and autumn 2016 to overall open-water period export of elements by WSL rivers, located in the permafrost-free (**A**) and permafrost-affected (**B**) regions.

### 3.2. Riverine Element Export Fluxes Across the Latitudinal Profile (Permafrost and Climate Gradient)

Because both concentrations [33,34,37] and seasonal fluxes (Supplementary Figure S1 and Table S1) of elements were not strongly impacted by the river size, the watershed-averaged fluxes can be calculated as a function of the watershed latitude. Furthermore, we used ternary molar diagrams (Ca − Mg − (Na + K) and Cl − SO$_4$ − HCO$_3$) to reveal the role of season, river size and permafrost coverage of major elementary composition (Supplementary Figure S3), following traditional geochemical classifications [54]. Regardless of the season, water chemical composition from permafrost-free rivers was distinctly different from water from permafrost-affected rivers and strongly enriched in Ca and HCO$_3^-$. As a result, the latitude (which corresponds with the permafrost zonation) can be considered as the main factor controlling major cation concentrations in the WSL rivers. The average (±2 s.d.) fluxes of dissolved (<0.45 μm) major element export from the WSL watersheds for each latitudinal belt in 2016 and 2015 are shown in Figure 3 and Supplementary Figure S4, respectively. For the sake of scientific rigor, we illustrate the yields separately for 2015 and 2016, but the two-year average values of element export during spring and summer are given in Table 1.

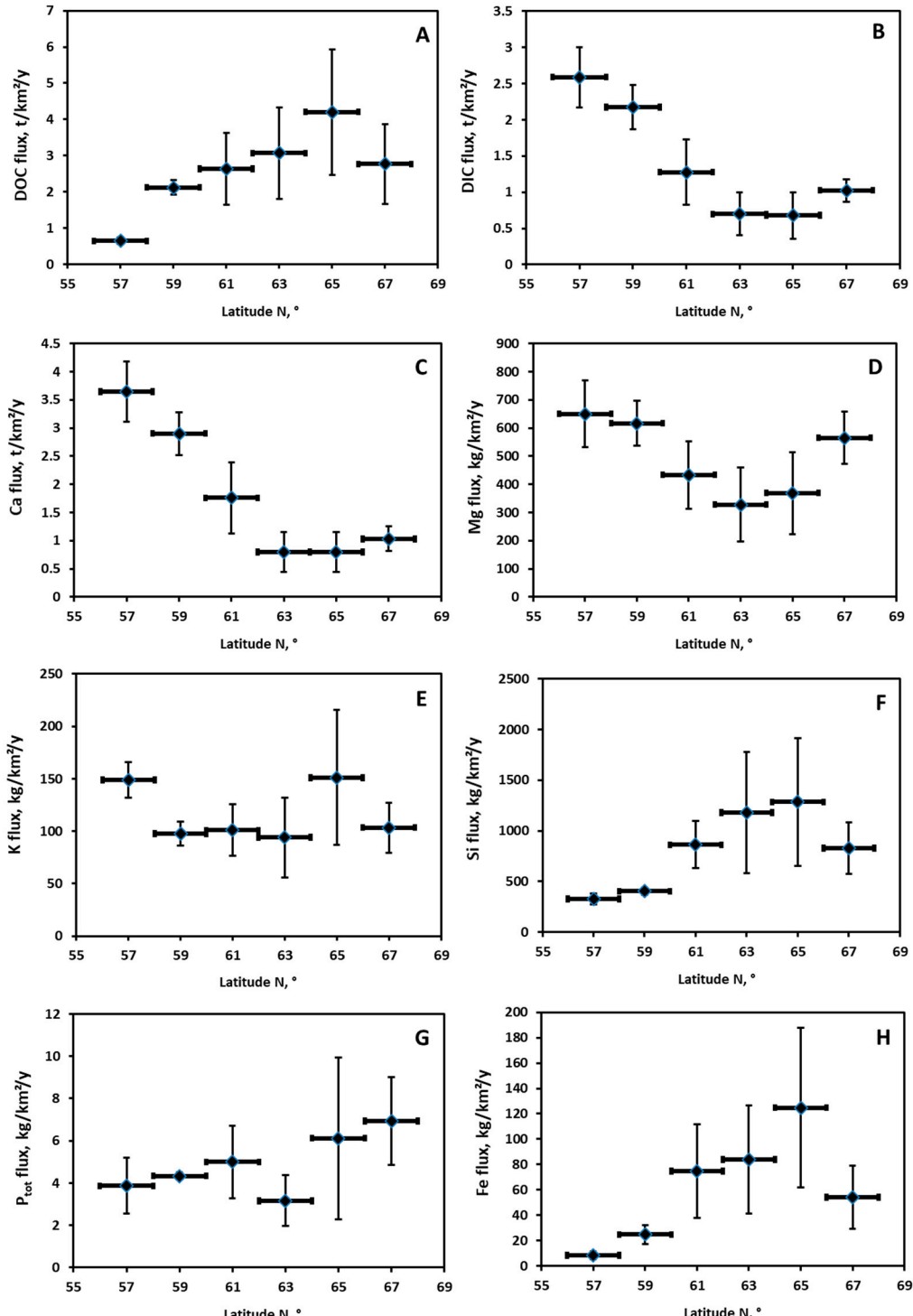

**Figure 3.** Yields (watershed-area normalized export fluxes) of DOC (**A**), DIC (**B**), Ca (**C**), Mg (**D**), K (**E**), Si (**F**), $P_{tot}$ (**G**) and Fe (**H**) during May-October 2016 in 33 WSL rivers across the latitudinal gradient. The vertical uncertainties represent the 2 s.d. of several rivers belonging to the same latitudinal belt. Thick horizontal bars represent the 2° latitudinal belts. The year is represented by open-water period (May to October), neglecting winter (November to April), when small rivers freeze solid in the north.

**Table 1.** Mean (±SD) export fluxes of major and trace elements by WSL rivers in the permafrost-free and the permafrost zone.

| Element kg/km$^2$/y | Biennial Flux, 2015 + 2016 (May—August) | | Annual Flux, 2016 (May—October) | |
|---|---|---|---|---|
| | **Permafrost-Free** | **Permafrost** | **Permafrost-Free** | **Permafrost** |
| **DOC** | 1192 ± 310 | 6340 ± 1858 | 1165 ± 202 | 3399 ± 1279 |
| *Major anions* | | | | |
| **DIC** | 2122 ± 389 | 1150 ± 316 | 2278 ± 403 | 839 ± 280 |
| **Cl** | 132 ± 50 | 2092 ± 1052 | 160 ± 84 | 1150 ± 692 |
| **SO$_4$** | 409 ± 201 | 308 ± 112 | 486 ± 319 | 234 ± 116 |
| *Major cations* | | | | |
| **Na** | 452 ± 115 | 1784 ± 769 | 528 ± 163 | 1116 ± 592 |
| **Mg** | 554 ± 103 | 615 ± 174 | 602 ± 107 | 414 ± 125 |
| **Ca** | 3045 ± 545 | 1585 ± 510 | 3125 ± 557 | 1004 ± 368 |
| **K** | 136 ± 21 | 175 ± 56 | 130 ± 20 | 108 ± 36 |
| *Macro-nutrients* | | | | |
| **P$_{tot}$** | 6.41 ± 2.4 | 22.4 ± 8.2 | 3.66 ± 1 | 5.71 ± 2.3 |
| **P-PO$_4$** | 1.99 ± 0.67 | 10.7 ± 3.8 | - | - |
| **N-NO$_3^-$** | 7.4 ± 1.8 | 8.12 ± 3.4 | - | - |
| **N-NH$_4^+$** | 3.57 ± 0.8 | 14.6 ± 6.1 | - | - |
| **Si** | 409 ± 73 | 1528 ± 415 | 426 ± 112 | 1076 ± 430 |
| *Micro-nutrients* | | | | |
| **B** | 2.26 ± 0.43 | 5.63 ± 2.1 | 2.23 ± 0.39 | 3.82 ± 1.8 |
| **Mn** | 3.46 ± 1.8 | 36.7 ± 13 | 1.4 ± 0.69 | 16.9 ± 11 |
| **Fe** | 32.9 ± 14.3 | 292.7 ± 93.8 | 17.3 ± 7.5 | 92.2 ± 42 |
| **Co** | 0.012 ± 0.005 | 0.15 ± 0.06 | 0.009 ± 0.002 | 0.066 ± 0.04 |
| **Ni** | 0.126 ± 0.03 | 0.599 ± 0.22 | 0.118 ± 0.018 | 0.322 ± 0.15 |
| **Cu** | 0.159 ± 0.06 | 0.283 ± 0.12 | 0.128 ± 0.041 | 0.177 ± 0.087 |
| **Zn** | 1.31 ± 1 | 4.55 ± 3 | 0.203 ± 0.091 | 1.42 ± 0.89 |
| **Rb** | 0.092 ± 0.016 | 0.261 ± 0.098 | 0.078 ± 0.014 | 0.159 ± 0.059 |
| **Mo** | 0.023 ± 0.01 | 0.011 ± 0.004 | 0.029 ± 0.017 | 0.006 ± 0.002 |
| **Ba** | 3.32 ± 0.52 | 7.89 ± 3.4 | 2.87 ± 0.37 | 2.95 ± 1.4 |
| *Toxicants* | | | | |
| **As** | 0.098 ± 0.021 | 0.337 ± 0.11 | 0.093 ± 0.026 | 0.143 ± 0.047 |
| **Cd** | 0.0009 ± 0.0004 | 0.005 ± 0.002 | 0.0008 ± 0.0003 | 0.003 ± 0.001 |
| **Sb** | 0.008 ± 0.003 | 0.016 ± 0.005 | 0.008 ± 0.003 | 0.009 ± 0.004 |
| **Pb** | 0.023 ± 0.009 | 0.065 ± 0.031 | 0.005 ± 0.002 | 0.024 ± 0.012 |
| *Geochemical traces* | | | | |
| **Li** | 0.193 ± 0.059 | 0.57 ± 0.17 | 0.272 ± 0.094 | 0.455 ± 0.18 |
| **Al** | 5.37 ± 2.8 | 41.5 ± 18 | 5.86 ± 2.6 | 20.4 ± 11 |
| **Ti** | 0.223 ± 0.075 | 1.58 ± 0.56 | 0.512 ± 0.1 | 1.4 ± 0.52 |
| **Cr** | 0.041 ± 0.01 | 0.379 ± 0.12 | 0.024 ± 0.009 | 0.204 ± 0.08 |
| **Ga** | 0.002 ± 0.001 | 0.007 ± 0.002 | 0.0009 ± 0.0004 | 0.004 ± 0.002 |
| **Sr** | 15.7 ± 3 | 17.2 ± 8.8 | 16.6 ± 3.4 | 9.12 ± 4.8 |
| **Y** | 0.015 ± 0.006 | 0.11 ± 0.05 | 0.013 ± 0.003 | 0.056 ± 0.03 |
| **La** | 0.036 ± 0.029 | 0.085 ± 0.035 | 0.012 ± 0.003 | 0.05 ± 0.026 |
| **Ce** | 0.059 ± 0.048 | 0.177 ± 0.082 | 0.02 ± 0.006 | 0.091 ± 0.051 |
| **Pr** | 0.003 ± 0.001 | 0.023 ± 0.01 | 0.003 ± 0.0008 | 0.012 ± 0.007 |
| **Nd** | 0.013 ± 0.006 | 0.096 ± 0.044 | 0.012 ± 0.003 | 0.049 ± 0.028 |
| **Sm** | 0.003 ± 0.001 | 0.022 ± 0.01 | 0.003 ± 0.0008 | 0.012 ± 0.006 |
| **Eu** | 0.0009 ± 0.0003 | 0.006 ± 0.002 | 0.001 ± 0.0002 | 0.003 ± 0.002 |
| **Tb** | 0.0006 ± 0.0005 | 0.012 ± 0.006 | 0.003 ± 0.0007 | 0.012 ± 0.007 |
| **Gd** | 0.003 ± 0.001 | 0.014 ± 0.006 | 0.0007 ± 0.0002 | 0.002 ± 0.0009 |
| **Dy** | 0.003 ± 0.001 | 0.02 ± 0.009 | 0.003 ± 0.0007 | 0.01 ± 0.006 |
| **Ho** | 0.0005 ± 0.0002 | 0.004 ± 0.002 | 0.0005 ± 0.0001 | 0.002 ± 0.001 |
| **Er** | 0.002 ± 0.0007 | 0.012 ± 0.005 | 0.001 ± 0.0004 | 0.006 ± 0.003 |
| **Tm** | 0.0002 ± 0.0001 | 0.002 ± 0.0007 | 0.0002 ± 0.00005 | 0.0009 ± 0.0005 |
| **Yb** | 0.001 ± 0.0006 | 0.011 ± 0.005 | 0.001 ± 0.0003 | 0.006 ± 0.003 |
| **Lu** | 0.0002 ± 0.0001 | 0.002 ± 0.0007 | 0.0002 ± 0.00005 | 0.0009 ± 0.0005 |

**Table 1.** *Cont.*

| Element kg/km$^2$/y | Biennial Flux, 2015 + 2016 (May—August) | | Annual Flux, 2016 (May—October) | |
|---|---|---|---|---|
| | Permafrost-Free | Permafrost | Permafrost-Free | Permafrost |
| Zr | 0.901 ± 0.33 | 1.47 ± 0.31 | 1.27 ± 0.46 | 1.57 ± 0.6 |
| Cs | 0.0003 ± 0.0001 | 0.0021 ± 0.0009 | 0.0003 ± 0.0001 | 0.001 ± 0.0005 |
| Nb | 0.0007 ± 0.0002 | 0.004 ± 0.002 | 0.0009 ± 0.0003 | 0.002 ± 0.001 |
| Hf | 0.0007 ± 0.0004 | 0.008 ± 0.003 | 0.001 ± 0.0004 | 0.005 ± 0.003 |
| Th | 0.002 ± 0.0009 | 0.009 ± 0.004 | 0.002 ± 0.0005 | 0.005 ± 0.003 |
| U | 0.021 ± 0.01 | 0.005 ± 0.002 | 0.03 ± 0.019 | 0.003 ± 0.001 |

Although the maximal uncertainties on discharge do not exceed 30% (see Section 2.2), the regional element yield assessments are subjected to high variation among rivers (Figure S1) with s.d. often at 50% (see Figure 3, Supplementary Figure S4). Among possible causes of these sizable uncertainties are non-linear and hysteretic relationships between concentrations and discharge which could not be resolved due to low frequency of sampling. Another reason of this variability could be highly dynamic behavior of element concentration during summer and autumn baseflow and spring flood, reflecting source limitation, chemostasis, or transport limitation (i.e., refs. [55–57]).

The DOC yield was minimal in rivers south to 59°N and exhibited a maximum at 61–65° N and 63–65° N in 2015 and 2016, respectively, with overall magnitude of variation by a factor of 4 (Figure S4A and Figure 3A, respectively). The yields of DIC and Ca decreased more than five-fold from south to north and achieved minimal values in the 63–65° N belt, for both years of observation (Figure 3B,C and Supplementary Figure S4B,C). Magnesium showed a weak minimum of yield in the 63–65° N which was however pronounced only in 2016; the overall variations were less than a factor of 2 (Figure 3D and Supplementary Figure S4D). Potassium yield remained fairly constant across the latitudinal profile with overall variations less than a factor of 1.5 to 2.0 between various latitudinal belts (Figure 3E and Supplementary Figure S4E). Silicon showed a three-fold increase in export fluxes from the south to the north, quite similar for two years of monitoring (Figure 3F and Supplementary Figure S4F).

The latitudinal pattern of other major and trace element export fluxes followed these three main types of behavior described above: (1) minimal in the south and a northward increase by a factor of 3 to 5 (Si, POC, P-PO$_4$; N-NH$_4$, Ni, REEs, Zr, Nb, Hf, and Th), with a local maximum at 63–65° N (DOC, Al, Ti, Cr, Mn, Fe, Co, Ni, Cu, Zn, Ga, Rb, Cd, Cs, and Pb); (2) northward decrease by a factor of 3 to 5 (DIC, Ca, SO$_4$, Sr, Mo, W, and U), and (3) overall independence of yield on latitude, or the latitudinal variations were less than a factor of 2 (Cl, N-NO$_3$, Li, B, Mg, P$_{tot}$, K, As, Zr, Sb, and Ba). Note that some elements (Mg, Zn, As, Rb) may belong to one or another group depending on the year of observation.

The yields of all 51 dissolved elements are listed in Table 1, which presents the export fluxes of two-year average (May to August) and open-water period (May to October 2016) in permafrost-free and permafrost-affected zones. The ratio of element flux in the non-permafrost zone to that in the permafrost zone (R$_{absent/permafrost}$, Figure 4) demonstrates a distinct order of elements whose export occurs preferentially in southern or northern part of the WSL. There are three main groups of elements - those exhibiting the highest (a factor of 2 to 10) yield in the permafrost-free zone (DIC, SO$_4$, Ca, Mo and U) and those showing a maximum in the permafrost-affected zone (As, P$_{tot}$, Li, B, Rb, Na, Si with $0.5 \leq$ R$_{absent/permafrost}$ $\leq 0.25$, and Ni, Th, Cd, DOC, Cs, Nb, Ti, Al, Fe, Co, Cr, Mn, Hf, Co, and REEs with R$_{absent/permafrost}$ <0.25). The elements of an intermediate group showed comparable (±30%) yields in permafrost-free and permafrost-affected parts of the WSL (Sr, Mg, K, Ba, Sb, Zr, Cu).

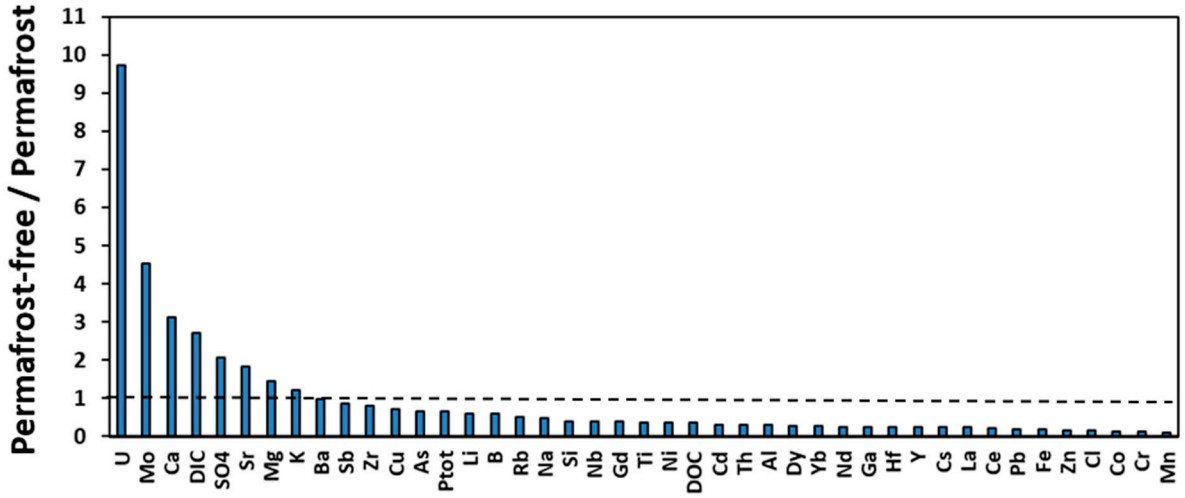

**Figure 4.** A histogram of the ratio of element average yield in the permafrost-free rivers to that in the permafrost-affected rivers for 6 open-water months (May to October) of 2016 (based on the data listed in Table 1).

## 4. Discussion

### 4.1. Effect of Seasons, Watershed Size, Latitude and Permafrost Coverage on Elementary Yields

There are several distinct groups of elements defined according to the latitudinal patterns of their yields. These groups reflect the relative mobility of elements, consistent with general knowledge of river hydrochemistry in the high latitude regions [58]. The DOC, organically-bound metals (V, Cr, Mn, Fe, Co, Ni, Cu, Zn, Cd and Pb) and many low soluble TE–geochemical traces (Al, REEs, Nb, Ti, Zr, Hf, and Th) exhibited similar 1st-type latitudinal pattern (minimum in the south and maximum in the permafrost-affected zone). The overall open-water period transport of these elements was dominated by spring flood, as it was also observed in other boreal and subarctic rivers (i.e., the Severnaya Dvina River, [59]). The nutrients (K, Si, P-PO$_4$, P$_{tot}$, and N-NH$_4^+$) and TE—analogous of macronutrients (Rb)—exhibited 50 ± 10% share of annual export during spring flood and a northward increase. Finally, the yield of geochemically mobile elements (DIC, Cl$^-$, SO$_4$, Li, B, Mg, Ca, Sr, Mo, and U) decreased northward, with less than 40–50% of overall contribution provided by spring flood period.

The division of elements into these groups depends on (1) the share of different seasons in overall 6-month open water period export, (2) the shape of the latitudinal pattern (Figure 3 and Figure S4), and (3) the difference in total 2-year averaged riverine yields between permafrost-free and permafrost-affected zones (Table 1, Figure 4). Such a distinction is consistent with two main factors controlling the element export (transport) in the WSL rivers. First, this is source-limitation, when the input of elements from soils to the river is controlled by river connectivity to deep and shallow groundwater reservoirs [27]. Due to the presence of carbonate mineral concretions in clay-silt bedrocks, especially in the south of the WSL [33], the groundwater is enriched in soluble elements such as alkalis and alkaline-earth metals, oxyanions and U(VI). The second is transport-limitation, when the export of an element is controlled by the availability of its carrier such as organic and organo-ferric, organo-aluminum colloids [30,34,45]. This factor reflects the superposition of surface source (topsoil, vegetation) providing organic colloids, and deep soil (mineral horizons), together with Fe(II)-rich groundwaters, providing soluble TE. In addition to these two main groups of elements, the mineral nutrients are not limited by transport and exhibit quite complex pattern which reflects their deep (groundwater, bedrock lithology) and surface (plant litter, atmospheric deposition) sources. These elements are strongly impacted by their biotic uptake in the river channel or seasonal release from decaying aquatic macrophytes, plankton and periphyton [37,60]. Various internal factors, operating in soils and riparian zone of the river, are capable of modifying the export of nutrients. For example,

low export fluxes of both total P (phosphate and organic P) and P-PO$_4$ in southern (56–60° N) latitudinal belt compared to northern, permafrost-affected rivers (see Figure 3G, Figure S4G) may be due P retention via adsorption onto and coprecipitation with Al, Fe hydroxides in the deeper part of soil profile [61,62] as well as P removal in the form of Ca phosphate minerals [63] occurring in the unfrozen mineral soils, exposed to surface fluids. Furthermore, P uptake by abundant terrestrial and aquatic vegetation is most pronounced in the south. In the north, the mineral soils are essentially frozen, and the requirements of aquatic and terrestrial biota for this nutrient are much lower [64].

The effect of watershed size on DOC, DIC, cations, and Si export fluxes was rather minor. This is at odds with strong river size control on element yields as it is known in temperate and mountainous regions [65,66]. In particular, small catchments of wetlands exhibit generally lower runoff than the medium and large rivers, and the runoff is one of the major controlling factors of chemical weathering and element export [46,67–70]. Season also played a secondary role in determining element yield pattern. However, sizable correlations of element fluxes with S$_{watershed}$ in summer and autumn, observed solely in the permafrost-free zone (Supplementary Table S1) can be explained by more pronounced impact of deep underground waters in large rivers compared to small ones. These waters typically contain a high concentration of soluble, labile elements (e.g., Cl, SO$_4$, Na, Mo, As) [33,34] but also Fe(II), those oxidation in the riparian and hyporheic zones create large amount of organo-ferric colloids [45] capable to provide enhanced concentrations of typically insoluble low mobile elements such as trivalent and tetravalent hydrolysates.

The latitude was revealed to be the primary governing parameter of elementary yields and clearly marked the difference between permafrost-free, discontinuous and continuous permafrost regions. The most northern regions of the WSL exhibited rather high yields of DOC, DIC, Si and cations. It is possible that in continuous permafrost zone of frozen peat bogs, the underlining mineral layer is protected by the permafrost. As a result, the active (seasonally unfrozen) layer is located within the organic layer which comprises live vegetation, plant litter, and upper peat layer. This organic matrix is extremely reactive, and capable of releasing sizable amount of DOC, P, Si, Ca and nutrients over very short periods of time during contact with surface waters [71,72]. Water temperature (between 4 and 25 °C) has only minor effect on C and element release from both thawed and frozen peat [72]. Therefore, even short-term contact of water with surface peat and vegetation is capable mobilizing sizable amount of DOC and nutrients, despite low temperature in the northern regions. In this regard, the elementary export by the WSL rivers is strongly controlled by dynamics of peat formation/decay across the territory. The particularity of the WSL is that, currently, this region is recovering from the last glaciation. As a result, the ecosystems are highly non-stationary: the peat actively accumulates in the south [73], while in the north, the frozen peat is subjected to thawing and degradation [74–76]. The uptake of elements from groundwater, river and forest tree litter by growing peat in southern mires counteracts with DOC and element release from thawing/degrading peat in the northern palsas. The elements affected by these processes are those that exhibit the highest concentration in peat relative to undelaying mineral (silt, sand) horizons. According to a previous assessment of elementary peat composition in the WSL [77], the peat is sizably enriched in C, (V, Cr), trivalent (TE$^{3+}$) and tetravalent (TE$^{4+}$) trace metals (Al, Y, REEs, Ti, Zr, Hf, Th) and U, Zn, Pb and depleted in highly mobile alkalis and alkaline-earths metals, As, and Mo. Therefore, enhanced riverine yield of low-soluble TE$^{3+}$, TE$^{4+}$, U, and some divalent metals in the north relative to the south is possibly due to these elements being tightly linked to peatland evolutionary pattern across the WSL. Similarly, depletion of peat relative to underlying mineral horizons in soluble, highly mobile elements is consistent with enhanced export of these elements by southern rivers, where the peat accumulation occurs.

*4.2. Comparison of Major Cation, DIC, Si and DOC Export Fluxes in the WSL with Other Boreal Regions*

The average total dissolved cation flux (TDS_c = Na + K + Ca + Mg) over May-October 2016 ranged from 4.40 ± 0.55 t km$^{-2}$ y$^{-1}$ in the permafrost-free zone to 2.64 ± 0.59 t km$^{-2}$ y$^{-1}$ in the permafrost-affected zone, which is lower than the fluxes of Central Siberian rivers of the same latitude,

draining basaltic rocks (5 to 8 t km$^{-2}$ y$^{-1}$, [78]), large Siberian rivers such as Yenisey and Lena (6.2 and 6.8 t km$^{-2}$ y$^{-1}$, respectively, [79]), the Ob River in its middle course (6.0 t km$^{-2}$ y$^{-1}$, [80]), and the permafrost-free Eurasian Arctic rivers draining sedimentary rocks (Sev. Dvina, 9.5 t km$^{-2}$ y$^{-1}$, [59]; Pechora, 6.6 t km$^{-2}$ y$^{-1}$, [79]). The TDS_c yield of permafrost-affected WSL rivers is, however, comparable with previous estimations of that in the middle-size Siberian rivers (2.8, 2.5, and 2.3 t km$^{-2}$ y$^{-1}$ for Kolyma, Indigirka and Anabar, respectively [79]). The main reason for relatively low cationic fluxes of small WSL rivers compared to other large rivers of Northern Eurasia of similar runoff are i) low connectivity of WSL rivers with underlying bedrocks and groundwater, due to thick peat layer and permafrost, and ii) essentially weathered character of silicate rocks (sands, clays) in western Siberia. Note also that the transport of major cations in permafrost-free rivers is strongly pronounced during winter (e.g., 35–40% of total annual yield in the Severnaya Dvina River [59]), so it is possible that cationic fluxes of WSL rivers during May-October are somewhat underestimated relative to annual export. At the same time, the majority of small (<4000 km$^2$ watershed area) rivers in the northern part of the WSL freeze solid during the winter [33], so the winter flux may be non-negligible only for southern, permafrost-free rivers.

In contrast to major cations, no difference in the export fluxes between small WSL rivers and large and medium size Eurasian rivers was detected for the DIC. The DIC export of small WSL rivers in the permafrost-free zone (2.2 ± 0.4 t km$^{-2}$ y$^{-1}$) is in agreement with recent estimations of DIC export in the middle course of the Ob River (2.9 t km$^{-2}$ y$^{-1}$, [80]) and with the mean riverine DIC yield of the entire Eurasian Arctic basin (2.2 t km$^{-2}$ y$^{-1}$, [79]). However, DIC export by permafrost-affected WSL rivers is somewhat lower (1–2 t km$^{-2}$ y$^{-1}$) and comparable to medium size rivers of Central and Eastern Siberia (0.6–2.2 t km$^{-2}$ y$^{-1}$, [79]). A tentative explanation of elevated DIC (but not TDS_c) flux in the WSL rivers is $CO_2$ and $HCO_3^-$ generation by microbial (and photolytic) processing of peat soil organic carbon (both DOC and POC), which is the main cause of very high $CO_2$ emission from WSL inland waters [81,82]. The light isotopic composition of DIC in the WSL rivers ($-25 \leq \delta^{13}C_{DIC} < -10$ ‰, [33]) is consistent with this hypothesis.

The riverine Si fluxes in the southern part (<61° N) of the WSL (0.5–1.0 t km$^{-2}$ y$^{-1}$) are comparable to those of the Ob River (0.62 t km$^{-2}$ y$^{-1}$, [80] and small rivers of the northern Sweden (0.9 t km$^{-2}$ y$^{-1}$, [83]). In contrast, the fluxes of the permafrost-affected rivers (1.0–1.5 t km$^{-2}$ y$^{-1}$, Figure 3F and Figure S4F and Table 1) contradict the expected trend of decreasing flux with the increasing latitude and decreasing temperature, given that the chemical weathering of silicate rocks is much slower in colder climates [67–70]. In fact, we noted that the northern fluxes of WSL are comparable with those of the temperate rivers such as Mississippi and Yangtze [84]. Moreover, if the silicate rock weathering significantly controls element delivery from the soil to the river, such a northward increase would occur for cations (Ca, Mg, Na) as it is known for typical silicate terrains of the boreal zone [85], but this is not observed in the WSL territory. As such, we hypothesize that two- to three-fold increase in Si yield of northern rivers relatively to southern rivers is linked to combination of (1) enhanced mobilization from plant litter via suprapermafrost flow in the permafrost zone, (2) limited silicate secondary mineral formation in shallow, essentially frozen northern soils compared to southern soils, and (3) strong uptake of Si by both terrestrial plants and aquatic macrophytes and periphyton in the southern rivers.

The DOC fluxes in the permafrost-affected regions of the WSL territory were quite high (2 to 6 t km$^{-2}$ y$^{-1}$) compared to the middle and lower reaches of the Ob River (0.64 and 1.2 t km$^{-2}$ y$^{-1}$, respectively, [79,80]). At the same time, these fluxes are comparable with those in large boreal river draining permafrost-free wetlands (the Severnaya Dvina River, 4.2 ± 0.8 t C$_{org}$ km$^{-2}$ y$^{-1}$, [59]) and are the highest among all known rivers flowing to the Arctic Ocean. Indeed, the Ob, Yukon, Lena, Yenisey, and Mackenzie rivers exhibit a DOC yield from 0.5 to 2.5 t km$^{-2}$ y$^{-1}$ ([79,86]). We suggest that the main factor responsible for such high DOC yield in small WSL rivers is high proportion of peatlands on their watershed (i.e., typically from 40 to 60%, according to the GIS data [34]). The peatland-draining Taz (watershed = 150,000 km$^2$), Pur (112,000 km$^2$), and Nadym (64,000 km$^2$) rivers, located entirely

in the discontinuous permafrost zone, also have a DOC yield of 1.9, 2.1, and 4.4 t $C_{org}$ km$^{-2}$ y$^{-1}$, respectively ([79] and calculations based on data from the RHS).

The northward increase in DOC flux possibly reflects strong leaching of OM from the plant litter and organic-rich topsoil (Histosol). In the north, the adsorption of DOM on underlying mineral horizons is minimal because these horizons are frozen. As a result, the riverine DOC export in the permafrost zone of the WSL is controlled by water travel time through the peat layer and underlying mineral horizons and the water residence time necessary for DOC leaching from upper vegetation layer (moss, lichen, litter). However, quantitative modeling of DOM and element reactive transport in the WSL peatlands, on the scale of a small watershed or large river, is beyond the scope of this study.

### 4.3. Possible Evolution of Western Siberia Rivers Elementary Yields Under Climate Change Scenario

The space for time approach employed in the present work provides some future projections for riverine element behavior. However, it exhibits a number of shortcomings whose analysis goes beyond the scope of this work. In particular, this approach does not address the time scale, necessary for the northern ecosystem to reach the new "more southern" state; it ignores possible shift in the structure of vegetation and soil microbial community that can indirectly impact the carbon and nutrient, in terms of landscape storage and removal via rivers, and it does not include information on the altered seasonality such as extended hydrologic seasons, earlier snowmelt, higher precipitation that will likely occur as a result of climate change. Nevertheless, as a first order empirical assessment, the following predictions can be made. The first consequence of climate warming in western Siberia is thawing of frozen peat and underlying mineral horizons. The thickness of the active layer (ALT) is projected to increase more than 30% during this century across the tundra area in the Northern Hemisphere [87–89]. In the WSL, this increase will be most dramatic in the north, where the peat deposits are thinner than those in the discontinuous permafrost zone [48,90–92]. The main consequences of the ALT increase may be the involvement of mineral (clay) horizons into water infiltration within the soil profile [90]. As a result, the DOC originated from the leaching of the upper peat layers and plant litter degradation will be retained on mineral surfaces via adsorption onto and incorporation into clay interlayers [93–97]. For inorganic solutes, the effect of ALT increase will be lower than that of DOC, given much lower affinity of $HCO_3$, cations and Si to clay surfaces and the lack of unweathered (primary) silicate rocks underneath the peat soil column. However, if the thawing will open new water paths between deep groundwater reservoirs and the river, this may increase the riverine export of major cations and DIC [9,33].

The second consequence of climate change in western Siberia is a shift of the permafrost zone boundaries further north [98–100]. Within the substituting space for time scenario, such a permafrost boundary change can be considered equivalent to the northward shift of the river latitudes as shown in Figure 3. Based on latitudinal pattern of major and TE (as illustrated in Figure 3 and Figure S4), the following groups of the river water components likely to change their riverine yield over open-water period in case of anticipated shift in the permafrost zones: (1) Elements those yields likely to increase by a factor of 2 to 3: DIC, Ca, $SO_4$, Sr, Ba, Mo, U; (2) Elements likely to decrease the yields by a factor of 2 to 5: DOC, Fe, Si, Ptot, P-$PO_4$, N-$NH_4$, divalent heavy metals, trivalent and tetravalent hydrolysates, and (3) Elements weakly affected by the permafrost boundary change and the ALT increase (change of yield by a factor of 1.5 to 2.0): B, Li, K, Rb, Cs, N-$NO_3$, Mg, Zn, As, Sb, Rb, Cs).

The impact of climate warming on riverine fluxes in the WSL is not restricted to the shift of permafrost zones and the change of water flow path (deep versus surface). It has to be placed in the context of changing precipitation, plant biomass productivity and modification of the seasonality [101]. Complex evaluation of these factors goes beyond the scope of this study and it requires ecosystem-level regional modeling. Overall, the WSL is likely to have increased lateral export of DIC but decreased export of DOC. However, the changes of both fluxes (+2 and −3 to −4 t C km$^{-2}$ y$^{-1}$ for DIC and DOC, respectively) are dwarfed compared to possible magnitude of C emission from WSL inland waters (rivers and lakes) to the atmosphere: 10–20 t km$^{-2}$ y$^{-1}$ in permafrost-free and isolated zone and 20–40 t C km$^{-2}$ y$^{-1}$ in discontinuous and continuous permafrost zone [102].

## 5. Conclusions

Based on a two-year seasonal sampling of 32 western Siberian rivers of various size (from 10 to $10^5$ km$^2$ in watershed area) draining through a sizable permafrost gradient, we measured riverine export fluxes of dissolved (<0.45 μm) C, nutrients, major and TE over a six-month open-water period (May to October). The export fluxes of DOC, DIC, major cations, macro- and micro-nutrients, toxicants, and geochemical tracers were weakly dependent on the size of the river. The primary parameter of export fluxes control was latitude, which marked the position of the permafrost zones.

There are several distinct groups of elements defined according to the latitudinal patterns of their yields. These groups reflect the relative mobility of elements, consistent with general knowledge of river hydrochemistry in high latitude regions. The DOC, organically-bound metals (V, Cr, Mn, Fe, Co, Ni, Cu, Zn, Cd and Pb) and many low soluble TE (Al, Ti, Zr, Hf, Th and REEs) exhibited similar latitudinal pattern with a minimum in the south and a maximum in the permafrost zone). A northward increase in Si export flux may be due to a decrease in Si uptake by plants in the north and strong Si retaining by mire and forest vegetation in the south.

An increase in DOC export fluxes from the south to the north (by a factor of 3 to 4 depending on the years of observation) could be due to leaching of OM from the plant litter and organic-rich topsoil (Histosol). The removal of DOC by adsorption on mineral horizons was hypothesized to be very low in the north. As a result, the riverine DOC export in the permafrost zone of the WSL may be strongly controlled by the water residence time necessary for DOC leaching from upper vegetation layer (moss, lichen, litter). This calls a need for quantitative modeling of DOM and element reactive transport in WSL peatlands, both at the scale of small watershed and large rivers. Furthermore, the peculiarity of western Siberia is that the elementary export by WSL rivers is strongly controlled by dynamics of peat formation and decay across the territory. Because the WSL is currently recovering from the last glaciation, this territory is dominated by non-stationary ecosystems with strong latitudinal contrast: the fresh peat is accumulating in the south whereas the old frozen peat is thawing and degrading in the north. As a result, uptake of elements from groundwater, river and forest tree litter by growing peat in southern bog competes with DOC and element release from thawing/degrading peat in the northern palsas. Note that, unlike many other permafrost-affected regions in the world whose $CO_2$ uptake rate during weathering is likely to increase under climate warming, the WSL may increase its riverine export of DIC but decrease the export of DOC.

**Supplementary Materials:** The following are available online at http://www.mdpi.com/2073-4441/12/6/1817/s1, Figure S1: Correlation between elementary fluxes and watershed area for permafrost-free and permafrost-affected regions; Figure S2: Partial contribution of spring, summer and autumn 2016 to overall open-water period export of anions and trace elements by WSL rivers; Figure S3: Ternary molar diagrams of major cations and anions in the WSL rivers. Figure S4: Yields of DOC (A), DIC (B), Ca (C), Mg (D), K (E), Si (F), $P_{tot}$ (G) and Fe (H) during May-August 2015 in 33 WSL rivers across the latitudinal gradient. Table S1: Spearman correlation coefficients ($p < 0.05$) between element export flux (yield) and watershed area.

**Author Contributions:** O.S.P. designed the study and wrote the paper; R.M.M., I.V.K., and S.V.L. performed sampling, analysis of cations and anions, and their interpretation; S.N.V. and S.N.K. were responsible for the choice of sampling objects and statistical treatment; S.V.L. provided the background information on soil, peat, and permafrost active layer; L.S.S. was in charge of DOC, DIC, and anion measurements and their interpretation; L.G.K. provided GIS-based interpretation, mapping, and identification of river watersheds; S.G.K. performed all primary hydrological data collection, and their analysis and interpretation. All 10 authors participated in field expeditions. All authors have read and agreed to the published version of the manuscript.

**Funding:** Russian Science Foundation: No 18-17-00237 and 18-77-10045. Russian Fund for Fundaental Research: 19-55-15002, 20-05-00729_a.

**Acknowledgments:** We acknowledge main support from RSF grant No 18-17-00237 and RFBR grants No 19-55-15002, 20-05-00729_a, and RSF grant No 18-77-10045 for field work.

**Conflicts of Interest:** The authors declare no conflict of interest.

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
