# Peer review of "Impact of Permafrost Thaw and Climate Warming on Riverine Export Fluxes of Carbon, Nutrients and Metals in Western Siberia"

_water, doi:10.3390/w12061817_

Round 1
Reviewer 1 Report
In the manuscript “Impact of permafrost thaw and climate warming on riverine export fluxes of carbon, nutrients and metals in western Siberia” Pokrovsky et al present results of very extensive survey of element concentrations by rivers in the West Siberian Lowlands and, by applying “substitution space for time” scenario, upscale the data to predict the effect of climate change on the fluxes.
My major concern with the manuscript is its novelty. In three earlier publication in Biogeochemistry (Pokrovsky et al 2015, 2016; Raudina et al 2017 and Krickov et al 2018), publications in Water (Vorobyev et al 2017), and Science of the Total Environment (Raudina et al, 2018), and two publications in BBA (Krykov et al 2019a, 2019b) the authors present data collected from the same survey and arrive at the same conclusion based on the same hypothetical scenario. I, therefore, encourage the authors to give special attention to the novelty of the reported findings and conclusions in light of their past publications.
Author Response
Reviewer #1:
Comments and Suggestions for Authors
In the manuscript “Impact of permafrost thaw and climate warming on riverine export fluxes of carbon, nutrients and metals in western Siberia” Pokrovsky et al present results of very extensive survey of element concentrations by rivers in the West Siberian Lowlands and, by applying “substitution space for time” scenario, upscale the data to predict the effect of climate change on the fluxes.
My major concern with the manuscript is its novelty. In three earlier publication in Biogeochemistry (Pokrovsky et al 2015, 2016; Raudina et al 2017 and Krickov et al 2018), publications in Water (Vorobyev et al 2017), and Science of the Total Environment (Raudina et al, 2018), and two publications in BBA (Krykov et al 2019a, 2019b) the authors present data collected from the same survey and arrive at the same conclusion based on the same hypothetical scenario. I, therefore, encourage the authors to give special attention to the novelty of the reported findings and conclusions in light of their past publications.
This is very pertinent comment, and we are well aware how crucial is the novelty issue. Fir of all, in no way did we intend to present the same data in different publications. All the text, figures and tables of the present manuscript are original, as can be also verified by search methods.
Previous work in the WSL rivers focused on major and trace element concentration based on sampling campaign of 2014 (Pokrovsky et al., 2015, 2016). The present paper is built on sampling performed in 2015 and 2016, and incorporates new original hydrological modeling to calculate element export fluxes. The export fluxes, were not at all considered in previous works of Pokrovsky et al (2015, 2016) on major and trace elements, Vorobyev et al (2017) on dissolved nutrients, and Krickov et al (2019, 2020) on colloids and particles. The main novelty of the present study is therefore in providing, for the first time, for such a large territory and such a high number of rivers, the region-specific exports fluxes (element yields). These “geochemical constants” are of high value for any further modeling of biogeochemical cycles of elements in the permafrost regions and as such represent high added value.
Finally, as for papers of Raudina et al. (2017, 2018), mentioned by the reviewer, these works deal with totally different aspects of WSL geochemistry - soil porewaters and supra-permafrost waters, and do not consider river concentration and fluxes.
In response to this remark, we added a short paragraph on the novelty of this work relative to previous study of our group in the Introduction (now lines 147-154).
Reviewer 2 Report
Review: Water Manuscript #805146
Summary
The study by Pokrovsky and others describe a study on how climate change via temperature increase and a shifting permafrost boundary will influence in the Western Siberia Lowland. They use a novel dataset from 32 rivers and a “space for time” repeated sampling approach to project how riverine fluxes of carbon, nutrients, cations, and metals will likely change at short and long timescales. The authors show using this gradient approach that fluxes of some materials (e.g., DOC, DIC, Si) are dependent on permafrost extent and latitude, while others reflected more conservative or dynamic behavior (e.g., K, Mg).
Overall, I thought this was an interesting and creative study demonstrating future impacts of climate change on understudied Arctic landscapes. The manuscript is certainly appropriate for this special issue, and their findings will be of interest to Water readers. However, I have several comments that should be addressed in a revision. I have included some comments below that I hope the authors find helpful.
Major Comments
While the substituting space for time is an interesting way to ask questions about the release of materials given warming temperatures and decreasing permafrost extent, the shortfall of this approach is that it assumes that all landscapes have the same base pools for given nutrients, metals, cations, etc. In other words, the space for time approach implies that a more northern location will directly become “more southern” as climate change progresses. While useful for providing some future projections for riverine behavior, it does not: 1) address how long it may take an ecosystem to reach the new “more southern” state, 2) consider different vegetation or soil community structure shifts that would less directly influence carbon, nutrient, etc. landscape pools (i.e., storage) and removal via rivers, nor 3) include information on the altered seasonality (i.e., extended hydrologic seasons, earlier snowmelt, etc.) that will likely occur as a result of climate change. While these effects may be beyond the sampling design of the authors, I suggest they touch on them more explicitly in their discussion.
Second, I have some concerns about how the error for calculating flux was incorporated into the authors’ analysis. Concentration-discharge relationships are often non-linear and hysteretic, such that a higher sampling frequency is necessary to estimate concentration at any given discharge. Additionally, concentrations are often most dynamic (i.e., different than baseflow patterns) at higher discharges – reflecting source limitation, chemostasis, or transport limitation behavior. Further, this error goes beyond the assigned percent error in calculating discharge. Therefore, the authors should take greater care to explain how their estimates are approximations that might not capture the full, dynamic hydrograph, as their analysis is not necessarily sufficient to reflect these patterns.
Lastly, in some parts of the discussion I found the authors’ references to be lacking. This was apparent in the section (starting at Line 397) how certain materials would be transported differently along permafrost gradients (e.g., Lines 426-434) and when authors discuss their expectations about watershed size on fluxes (e.g., Lines 435-442). While I only noted this for several sections of the paper, I just want to suggest that when the authors are explaining the reasons behind their findings, they take greater care to include references to their points.
Minor Comments
There are several paragraphs (e.g., the one that starts at Line 91) that are very short, and could be incorporated into the preceding paragraph.
A very minor stylistic suggestion, but throughout the paper the authors use parenthetical statements, where this information could be incorporated into the sentence more directly. For example, the sentence starting at Line 38 could be written as: “Permafrost peatlands cover roughly 2.8 million km2 or 14% of permafrost-affected areas, mostly in Northern Eurasia…and contain a huge amount of highly vulnerable carbon…”. Again, this is simply a suggestion and I leave this to the authors’ discretion.
I encourage the authors to reconsider the language of “spring”, “summer”, and “autumn”. While these are applicable to more temperate ecosystems, they are less descriptive for Arctic and subarctic temporal patterns. Please see Ernakovitch et al. 2014 for more details on defining Arctic seasons.
The authors might also find this resource useful in describing the lack of hydrologic data in the Arctic: Shiklomanov et al. 2002.
The authors should provide additional details on why the latitude class bins were selected.
Line-by-line
Line 14: “Lateral” should be changed to simply “riverine”. Lateral implies hillslope or runoff processes in rivers, while longitudinal describes upstream to downstream fluxes.
Line 42: “A huge amount”
Line 73: Change to “the study of”
Line 97: Change to “In particular”
Line 120: Change to “capable of delivering”
Line 126: What do you mean “to their current level”?
Sentence at line 142: Please include a citation here
Line 209: Should this be: “The two sampled years contrasted”?
Line 214: “prior to 2000”
Line 245: “used either an analogous river approach…”
Line 299: Typo - “fluxesin”
Lines 375-387: There were several elements still in parentheses that should be fixed
Figures & Tables:
I appreciated the detail of Figure 1, but wondered if the DOC in Spring, Summer, and Autumn could instead be presented as an inset graph. This would allow the reader to follow from South to North while preserving the vegetation and land cover information necessary to understand the sampling design.
It looks like the titles of Figure 2 are cut off and need to be fixed / omitted.
I am not sure why there are horizontal error bars on Figure 3, as the x-axes were defined as categorical bins.
Figure 4 should have a reference line at 1
Author Response
Reviewer No 2
The study by Pokrovsky and others describe a study on how climate change via temperature increase and a shifting permafrost boundary will influence in the Western Siberia Lowland. They use a novel dataset from 32 rivers and a “space for time” repeated sampling approach to project how riverine fluxes of carbon, nutrients, cations, and metals will likely change at short and long timescales. The authors show using this gradient approach that fluxes of some materials (e.g., DOC, DIC, Si) are dependent on permafrost extent and latitude, while others reflected more conservative or dynamic behavior (e.g., K, Mg).
Overall, I thought this was an interesting and creative study demonstrating future impacts of climate change on understudied Arctic landscapes. The manuscript is certainly appropriate for this special issue, and their findings will be of interest to Water readers. However, I have several comments that should be addressed in a revision. I have included some comments below that I hope the authors find helpful.
We thank the reviewer for overall positive evaluation of our work and addressed all comments below.
Major Comments
While the substituting space for time is an interesting way to ask questions about the release of materials given warming temperatures and decreasing permafrost extent, the shortfall of this approach is that it assumes that all landscapes have the same base pools for given nutrients, metals, cations, etc. In other words, the space for time approach implies that a more northern location will directly become “more southern” as climate change progresses. While useful for providing some future projections for riverine behavior, it does not: 1) address how long it may take an ecosystem to reach the new “more southern” state, 2) consider different vegetation or soil community structure shifts that would less directly influence carbon, nutrient, etc. landscape pools (i.e., storage) and removal via rivers, nor 3) include information on the altered seasonality (i.e., extended hydrologic seasons, earlier snowmelt, etc.) that will likely occur as a result of climate change. While these effects may be beyond the sampling design of the authors, I suggest they touch on them more explicitly in their discussion.
This is very important remark. We totally agree on possible shortcomings of the “space for time” approach and alerted the readers of possible biases, mentioned by the reviewer. However we do not feel competent enough to develop the evaluation of possible effects, and it was not within the objectives of this work, which is already on the long side. We thank the reviewer for bringing these issues out and revised the Discussion accordingly (section 4.3, L 483-491).
Second, I have some concerns about how the error for calculating flux was incorporated into the authors’ analysis. Concentration-discharge relationships are often non-linear and hysteretic, such that a higher sampling frequency is necessary to estimate concentration at any given discharge. Additionally, concentrations are often most dynamic (i.e., different than baseflow patterns) at higher discharges – reflecting source limitation, chemostasis, or transport limitation behavior. Further, this error goes beyond the assigned percent error in calculating discharge. Therefore, the authors should take greater care to explain how their estimates are approximations that might not capture the full, dynamic hydrograph, as their analysis is not necessarily sufficient to reflect these patterns.
This is also an insightful comment. We are aware of sizable uncertainties associated with our discharge and flux evaluation, as stated in the text (L264-276 of the 1st version, section 2.2). The reviewer listed a number of other possible biases, and we incorporated these possibilities in the revised text (now L 293-299 of section 3.2). We would like to point out, that, although the uncertainties on discharges do not exceed 30%, the regional- element yield assessments are subjected to high dispersity among different rivers (Fig. S1) and often achieve 50% (see the s.d. ranges in Figs. 3, S3). We alerted the reader about these uncertainties accordingly (L 295, section 3.2 and added necessary references, L299).
Lastly, in some parts of the discussion I found the authors’ references to be lacking. This was apparent in the section (starting at Line 397) how certain materials would be transported differently along permafrost gradients (e.g., Lines 426-434) and when authors discuss their expectations about watershed size on fluxes (e.g., Lines 435-442). While I only noted this for several sections of the paper, I just want to suggest that when the authors are explaining the reasons behind their findings, they take greater care to include references to their points.
This is very pertinent remark, and we added a big deal of missing references (about 14 in total) in all sections of the Discussion.
Minor Comments
There are several paragraphs (e.g., the one that starts at Line 91) that are very short, and could be incorporated into the preceding paragraph.
We followed this recommendation and combined several paragraphs together, in the Introduction, Methods (section 2.2), and Discussion (section 4.3).
A very minor stylistic suggestion, but throughout the paper the authors use parenthetical statements, where this information could be incorporated into the sentence more directly. For example, the sentence starting at Line 38 could be written as: “Permafrost peatlands cover roughly 2.8 million km2 or 14% of permafrost-affected areas, mostly in Northern Eurasia…and contain a huge amount of highly vulnerable carbon…”. Again, this is simply a suggestion and I leave this to the authors’ discretion.
We agree and revised the text accordingly.
I encourage the authors to reconsider the language of “spring”, “summer”, and “autumn”. While these are applicable to more temperate ecosystems, they are less descriptive for Arctic and subarctic temporal patterns. Please see Ernakovitch et al. 2014 for more details on defining Arctic seasons.
This is a delicate issue. Half of the considered rivers are located in permafrost-free taiga biome, where three main open-water seasons are fully pronounced. Only 10 most northern rivers from the tundra biome (continuous permafrost zone) exhibit weak contrast between spring (June-July), summer (August) and autumn. For consistency with previous works in this region and to maintain uniformity of season attribution across the latitudinal gradient, we would like to keep the conventional terms in the present manuscript. We thank the reviewer for pointing out very useful work of Ernakovich et al (2014) and we cited it in the revised manuscript (L516).
The authors might also find this resource useful in describing the lack of hydrologic data in the Arctic: Shiklomanov et al. 2002.
This is also very useful work, and we referred to it in the revised Introduction (L 66- 68).
The authors should provide additional details on why the latitude class bins were selected. These latitude class bins were selected based on : i) the permafrost map of the WSL, where the permafrost zones roughly follow the latitude, and ii) necessity to integrate sufficient number of rivers in each permafrost belt and latitudinal grid, to provide robust statistics; added to L 235-239 of the revised version.
Line-by-line
Line 14: “Lateral” should be changed to simply “riverine”. Lateral implies hillslope or runoff processes in rivers, while longitudinal describes upstream to downstream fluxes.
We agree and corrected as recommended
Line 42: “A huge amount” - changed
Line 73: Change to “the study of” - fixed
Line 97: Change to “In particular” - fixed
Line 120: Change to “capable of delivering” - changed
Line 126: What do you mean “to their current level”? - To the values observed in this region at the present time; corrected.
Sentence at line 142: Please include a citation here
We removed this paragraph from the revised version.
Line 209: Should this be: “The two sampled years contrasted”?
Agree and revised
Line 214: “prior to 2000” - fixed
Line 245: “used either an analogous river approach…” - corrected
Line 299: Typo - “fluxesin” - fixed
Lines 375-387: There were several elements still in parentheses that should be fixed
We removed parentheses for these elements
Figures & Tables:
I appreciated the detail of Figure 1, but wondered if the DOC in Spring, Summer, and Autumn could instead be presented as an inset graph. This would allow the reader to follow from South to North while preserving the vegetation and land cover information necessary to understand the sampling design.
We totally agree and revised this figure as requested, while presenting the DOC histograms in an insert graph.
It looks like the titles of Figure 2 are cut off and need to be fixed / omitted. We corrected this figure; thanks for catching this!
I am not sure why there are horizontal error bars on Figure 3, as the x-axes were defined as categorical bins.
Thick horizontal bars on this figure mark the latitudinal bins; we added this explanation in the revised figure caption.
Figure 4 should have a reference line at 1
We agree and added a dashed line.
We thank reviewer No 2 for very insightful and constructive remarks.
Reviewer 3 Report
The authors undertook the task of quantifying carbon, nutrients, and metal fluxes in permafrost free to permafrost continuous watershed in the Western Siberia Peatlands.
I found the study well-executed and the manuscript reporting clearly major findings originating from the study.
I have but minor suggestions on how the report might be improved:
- line 40: mark clearly power 106 ==> 106
- line 43: N ==> Northern
- line 97: In Particularly ==> In particular
Please notice that there are other small grammar and syntax errors throughout the text that should be corrected, best by a native English speaking person
Sampling:
- it is obvious that seasonal worming moves northward at a certain velocity. Sampling took several weeks. I wonder if it was possible that the 'same' water was sampled first up-river, then down-river. One way to deal with the problem would be sampling in the N-S general direction
- samples preservation ==> please describe, if any
- in the case of certain trace metals, concentrations must have been really low- most likely close or even not exceeding to the detection limits ==> please comment on the detection limits of particular metals (e.g. cadmium) and the additional uncertainty caused by concentrations not exceeding DL.
Discussion:
- please discuss how long it might take to return to the former elements fluxes after a temporal increase due to permafrost thawing
Author Response
Reviewer No 3
The authors undertook the task of quantifying carbon, nutrients, and metal fluxes in permafrost free to permafrost continuous watershed in the Western Siberia Peatlands.
I found the study well-executed and the manuscript reporting clearly major findings originating from the study.
We appreciate positive evaluation of our work.
I have but minor suggestions on how the report might be improved:
- line 40: mark clearly power 106 ==> 106 - Corrected
- line 43: N ==> Northern - Changed as requested
- line 97: In Particularly ==> In particular - Fixed
Please notice that there are other small grammar and syntax errors throughout the text that should be corrected, best by a native English speaking person
We performed all necessary spelling check and grammar correction
Sampling:
- it is obvious that seasonal worming moves northward at a certain velocity. Sampling took several weeks. I wonder if it was possible that the 'same' water was sampled first up-river, then down-river. One way to deal with the problem would be sampling in the N-S general direction
We followed the change of seasons during our sampling campaign and moved from south to north in spring and from north to south in autumn thus collecting the river water at approximately the same period of the discharge. We added this useful information in the revised text (L 168-171 of section 2.1). Actually, we always sampled the same river 2-3 times a year, but not the same water, as inquired by reviewer. The latter would be possible if we stayed on a boat which moved over the river, but we actually crossed different rivers by car.
- samples preservation ==> please describe, if any
We added the following: “Samples for DOC, DIC, major and trace elements were stored in the refrigerator during 1-2 months prior the analyses, while the samples for nutrients were kept frozen”, L 185-186, section 2.1)
- in the case of certain trace metals, concentrations must have been really low- most likely close or even not exceeding to the detection limits ==> please comment on the detection limits of particular metals (e.g. cadmium) and the additional uncertainty caused by concentrations not exceeding DL.
This is an important remark, and we referenced necessary works where these issues are discussed (L 190-195 of the revised text). For all major and most trace elements, analyzed by ICP MS, the concentrations in the blanks were below analytical detection limits (≤ 0.1-1 ng/L for Cd, Ba, Y, Zr, Nb, REE, Hf, Pb, Th, U; 1 ng/L for Ga, Ge, Rb, Sr, Sb; ~ 10 ng/L for Ti, V, Cr, Mn, Fe, Co, Ni, Cu, Zn, As). The international geostandard SLRS-5 (Riverine Water Reference Material for Trace Metals) was used to check the validity and reproducibility of analysis. All certified major (Ca, Mg, K, Na, Si) and trace elements (Al, As, B, Ba, Co, Cr, Cu, Fe, Ga, Li, Mn, Mo, Ni, Pb, all REEs, Sb, Sr, Th, Ti, U, V, Zn) concentrations of the SLRS-5 standard and the measured concentrations agreed with an uncertainty of 10–20%. We prefer to avoid repetition of the information provided in other published papers.
Discussion:
- please discuss how long it might take to return to the former elements fluxes after a temporal increase due to permafrost thawing
This is very insightful comment. Here we only foresee the short-term (first decades) and long-term (100-300 years) consequences of permafrost thaw. Therefore, we estimate the time of return on the scale of several hundred years. However, as it is stated in the text, because the WSL is recovering from the last glaciation, the ecosystems are highly non-stationary and characterized by contemporary peat accumulation in the south and frozen peat thawing/degradation in the north (L 405-408). As such we would like to avoid unsupported speculations given high complexity of the ecosystem response to on-going warming and intrinsic limitation of space for time approach. We also added to the revised text: “In particular, this approach does not address the time scale, necessary for the northern ecosystem to reach the new “more southern” state; it ignores possible shift in structure of vegetation and soil microbial community that can indirectly impact the carbon and nutrient, in terms of landscape storage and removal via rivers, and it does not include information on the altered seasonality such as extended hydrologic seasons, earlier snowmelt, higher precipitation that will likely occur as a result of climate change.” (L 483-491, section 4.3).
We thank the reviewer for very insightful remarks and corrections.
Reviewer 4 Report
Water
Manuscript Number: 805146
Title: Impact of permafrost thaw and climate warming on riverine export fluxes of carbon, nutrients and metals in western Siberia
Article Type: Research Paper
Keywords: river flux, weathering, organic matter, permafrost, trace element, river
This work is aimed at filling to quantifying the export fluxes of 30 Western Siberia Lowland rivers of various size, combined with new hydrological modeling of region and season-specific river runoff of the Western Siberia Lowland territory.
I recommend it for publication after a minor revision.
The main trouble points are:
Comments
Introduction
The introduction is too long, it is recommended to summarize
Study site and methods
The geological map is missing! I think a schematic one should be included
Very Important:
Although the concentrations of major and trace elements in WSL river waters were described in [33,34,54]. I think a geochemical classification should be inserted with at least triangular diagrams and a TIS, see for example:
Critelli, T., Vespasiano, G., Apollaro, C., Muto, F., Marini, L., & De Rosa, R. (2015). Hydrogeochemical study of an ophiolitic aquifer: a case study of Lago (Southern Italy, Calabria). Environmental earth sciences, 74(1), 533-543.
References
Add these references:
Critelli, T., Vespasiano, G., Apollaro, C., Muto, F., Marini, L., & De Rosa, R. (2015). Hydrogeochemical study of an ophiolitic aquifer: a case study of Lago (Southern Italy, Calabria). Environmental earth sciences, 74(1), 533-543.
Author Response
Reviewer No 4
This work is aimed at filling to quantifying the export fluxes of 30 Western Siberia Lowland rivers of various size, combined with new hydrological modeling of region and season-specific river runoff of the Western Siberia Lowland territory. I recommend it for publication after a minor revision.
We thank the reviewer for his/her positive evaluation.
The main trouble points are:
The introduction is too long, it is recommended to summarize
We greatly revised and shortened the Introduction, also avoiding the plagiarism issue.
Study site and methods
The geological map is missing! I think a schematic one should be included.
Following this valuable advice, we revised Fig. 1 and added lithology of Quaternary deposits.
Very Important:
Although the concentrations of major and trace elements in WSL river waters were described in [33,34,54]. I think a geochemical classification should be inserted with at least triangular diagrams and a TIS, see for example: Critelli, T., Vespasiano, G., Apollaro, C., Muto, F., Marini, L., & De Rosa, R. (2015). Hydrogeochemical study of an ophiolitic aquifer: a case study of Lago (Southern Italy, Calabria). Environmental earth sciences, 74(1), 533-543.
We only partially agree with this comment. The work us about element fluxes, not concentrations. By its design, our study does not present elementary concentrations in WSL rivers, and does not discuss them in the context of climate change. However, we added two ternary plots of rivers in section 3.2 together with a pertinent reference (new Fig. 3). This allowed to justify the strategy of watershed-averaged fluxes and test the effect of permafrost coverage and seasons; L 278-284 of the revised text. We thank the reviewer for this suggestion!
We thank anonymous reviewer for his/her thorough and constructive review.
Round 2
Reviewer 1 Report
General remarks:
Stating the specific focus of this draft and amending the Introduction section accordingly has been essential. Thank you. However the following sections should also be amended to keep this focus.
The draft requires major revision to bring the central findings and conclusions forward.
In the process of revision please harmonize terminology through the whole text, figures, figure legends and supplementary materials.
Below are some comments and suggestions to help address the above. However, in order to streamline and focus the text, major revision rather than specific point amendments shold be performed.
Comments:
Your title states that your study focuses on “carbon, nutrients and metals” but in the text your section sub-titles you have “elements”, “groups of elements” etc.
Intro is still too long – needs to be focused on specific questions addressed in the study, with no side-track information.
For example, the whole section between lines 94-108 deals with CO2 emissions as an example of the space-for-time approach, but is not relevant to the results presented below. Discussion of the approach is essential, especially when it seems to be still controversial. However, it should focus more on fundamental reasoning and less on previous applications.
Define OC, TE and other abbreviations used.
I would suggest rephrasing the last paragraph of the Introduction, showcasing your past publications as a step-ladder for the presented analyses (rather than mentioning its originality), as you do in Methods.
A single use of non-defined terms, e.g. “geochemical constants” is discouraged.
In the Methods section you mention:
LL 163-4: “The river runoff ranges from 190±30 mm y-1 in the south to 300±20 mm y-1 in the north [49]”
Because accessibility of the referenced study to “Water” audience is limited, please clarify whether seasonal runoff values are reported by Nikitin & Zemtsov and if yes – compare with your estimates.
L 174: replace “It is known” with “It is agreed” or "accepted"
L 176: years
L 181: The normal values here are defined as prior to 2000 - averages? then from X to 2000. Source?
L 183: cellulose acetate
L 185: how did you handle samples at field? cold packs? dry ice? cooler? ambient? for how long?
L 194: to check the correctness = validate?
Figure 1: What is your reason for presenting DOC on the map? You do not specifically refer to it in your text. The Google map should be mentioned.
Please be specific with terms: DOC is not an element.
Define any compound species you mention – Ptot etc.
LL 244-5: “The yields of DIC, Ca, DOC, Mg, Si, Ptot, K and Fe are illustrated in Fig. S1 of the Supplementary Information.” Please provide reason for combining the above into one bin.
In your Into you state: “here we assess for the first time, element export fluxes (yields)…”. You present “The yields of DIC, Ca, DOC, Mg, Si, Ptot, K and Fe are illustrated in Fig. S1 of the Supplementary Information”. Your Fig. 2 deals with “export of elements” – is this the same as yield? or flux?
In figures included in the main text please present findings directly relevant to posed questions and use the same terms.
If Fig. 3 mainly presents methodological details or supports your decision regarding methodological approaches, it could be moved to Supplementary. Also, please fix the overlay.
Table 1: What is “Absent”? What is the order of presented elements/compounds? How does the table present the results described in text? You could use shading to bin elements/compounds to groups or to highlight “elements (Mg, Zn, As, Rb) may belong to one or another group depending on the year of observation”.
Fig. 5. What is “Absent”?
L. 335: groups
In Supplementary Info: Please make sure all data/text are visible.
Author Response
Reviewer #1:
General remarks:
Stating the specific focus of this draft and amending the Introduction section accordingly has been essential. Thank you. However the following sections should also be amended to keep this focus.
The draft requires major revision to bring the central findings and conclusions forward.
We extended the Abstract and revised the Conclusions; we also shortened the Introduction and made it more focused on our main and novel objectives as recommended.
In the process of revision please harmonize terminology through the whole text, figures, figure legends and supplementary materials. We verified the terminology, revised figure legends and captions.
Below are some comments and suggestions to help address the above. However, in order to streamline and focus the text, major revision rather than specific point amendments should be performed. We performed reorganization of result presentation as recommended.
Comments:
Your title states that your study focuses on “carbon, nutrients and metals” but in the text your section sub-titles you have “elements”, “groups of elements” etc. We agree and verified throughout the text. “Elements” is a general term used for carbon, nutrients and trace metals.
Intro is still too long – needs to be focused on specific questions addressed in the study, with no side-track information.
For example, the whole section between lines 94-108 deals with CO2 emissions as an example of the space-for-time approach, but is not relevant to the results presented below. Discussion of the approach is essential, especially when it seems to be still controversial. However, it should focus more on fundamental reasoning and less on previous applications.
We removed a big deal of the text in L 94-108 and in L 108-118; this allowed greatly condense and focus the Introduction. Note that the goal of this work was not to resolve the existing controversies, but to provide solid data on riverine export fluxes. These data are currently not available and they are essential for further biogeochemical modeling.
Define OC, TE and other abbreviations used. Good point, we carefully verified and defined where necessary.
I would suggest rephrasing the last paragraph of the Introduction, showcasing your past publications as a step-ladder for the presented analyses (rather than mentioning its originality), as you do in Methods. We totally agree and revised as following: “Based on our previous studies of the WSL rivers [29-31, 33, 34, 37], here we assess for the first time, element export fluxes (yields) across a large permafrost-affected territory and large number of rivers. The obtained elementary yields are essential for further modeling in biogeochemical cycles of elements in the permafrost regions.”
A single use of non-defined terms, e.g. “geochemical constants” is discouraged. Agree and replaced by “elementary yields”.
In the Methods section you mention:
LL 163-4: “The river runoff ranges from 190±30 mm y-1 in the south to 300±20 mm y-1 in the north [49]”. Because accessibility of the referenced study to “Water” audience is limited, please clarify whether seasonal runoff values are reported by Nikitin & Zemtsov and if yes – compare with your estimates. These are annual runoff values and they are used to characterize the general setting of the territory. Nikitin and Zemtsov did not present seasonal runoff. However, in our estimates for small and medium rivers of palsa and polygonal bogs of the permafrost zone, we used empirical equations accounting for measured hydrological parameters of these watersheds [Novikov et al., 2009].
L 174: replace “It is known” with “It is agreed” or "accepted" - revised as “accepted”
L 176: years - Fixed.
L 181: The normal values here are defined as prior to 2000 - averages? then from X to 2000. Source? These are mean multi-annual data from 1970 to 2000. We provided the reference to Roshydromet (ref. 44 of the revised version)
L 183: cellulose acetate - corrected accordingly
L 185: how did you handle samples at field? cold packs? dry ice? cooler? ambient? for how long? Immediately after filtration, samples were stored in the refrigerator, added to the text. Note that all our expedition vehicles are equipped with refrigerators.
L 194: to check the correctness = validate? Yes, to validate the analyses; corrected.
Figure 1: What is your reason for presenting DOC on the map? You do not specifically refer to it in your text. The Google map should be mentioned. The seasonal and spatial pattern of DOC export (yield) is certainly the most important finding of this work, and we thought it will be useful to illustrate it in the map.
Please be specific with terms: DOC is not an element. We agree and revised throughout the text. Actually, we always present DOC and other major and trace elements separately. This is common in the biogeochemical literature.
Define any compound species you mention – Ptot etc. We carefully checked the text for relevant definitions
LL 244-5: “The yields of DIC, Ca, DOC, Mg, Si, Ptot, K and Fe are illustrated in Fig. S1 of the Supplementary Information.” Please provide reason for combining the above into one bin. We explained the context and justification of this presentation as following: “The yields of DOC, representative major solutes (DIC, Ca, Mg) and nutrients (Si, Ptot, K, Fe)…”
In your Into you state: “here we assess for the first time, element export fluxes (yields)…”. You present “The yields of DIC, Ca, DOC, Mg, Si, Ptot, K and Fe are illustrated in Fig. S1 of the Supplementary Information”. Your Fig. 2 deals with “export of elements” – is this the same as yield? or flux? Yes, we used the general term “export fluxes” as “yields”. Corrected in Fig. 2.
In figures included in the main text please present findings directly relevant to posed questions and use the same terms. We verified the figures content and the terms accordingly.
If Fig. 3 mainly presents methodological details or supports your decision regarding methodological approaches, it could be moved to Supplementary. Also, please fix the overlay. We agree and moved it to the Supplement. We revised the labelling which was not visible in the first version. Note that, given large number of rivers, some overlay in the data points is inevitable.
Table 1: What is “Absent”? What is the order of presented elements/compounds? How does the table present the results described in text? You could use shading to bin elements/compounds to groups or to highlight “elements (Mg, Zn, As, Rb) may belong to one or another group depending on the year of observation”. In response to this remark, we reorganized this table to groups of elements: DOC, major anions (DIC, Cl, SO4), major cations (Na, Mg, Ca, K), macronutrients (P, N, Si), micronutrients (B, Mn, Fe, Co, Ni, Cu, Zn, Rb, Mo, Ba), toxicants (As, Cd, Sb, Pb) and geochemical tracers (Li, Al, Ti, Cr, Ga, Y, REEs, Zr, Hf, Th, U). We also explained the terms in the Table heading.
Fig. 5. What is “Absent”? Permafrost-free, revised.
- 335: groups - Fixed.
In Supplementary Info: Please make sure all data/text are visible. - Revised and corrected as necessary.
We thank the reviewer for very constructive review.
Reviewer 2 Report
I would like to thank the authors for taking such care to respond to my concerns. The manuscript is much improved and I am satisfied with the changes.
Author Response
We thank the reviewer for positive evaluation of our work.
Round 3
Reviewer 1 Report
After two rounds of revision, the study by Pokrovsky et al has been edited for clarity of presentation and can be accepted for publication after fully addressing the two remaining points.
1. Intermittent use of absent/permafrost-free and permafrost-bearing/ permafrost-affected. If you refer to different area definition – clarify. If the term couples refer to the same area – please chose one and use it exclusively through your whole text, figures, tables and supplementary materials. Your text is already heavy on its content, so why complicate its reading even further?
2. Proper reference to your “most important finding" presented in Figure 1 is still missing. In your response you wrote: "The seasonal and spatial pattern of DOC export (yield) is certainly the most important finding of this work, and we thought it will be useful to illustrate it in the map." Why don't you refer reader to it from your results section?
Minor suggestions:
L 120: Building upon?
L 156: Immediately after filtration, samples for DOC, DIC, major and trace elements were refrigerated and stored at +4C for 1-2 months prior to the analyses.
Fig. 1 legend: Google maps (not map)
4.1 Role of… in = Effect of … on..?
Author Response
Reviewer #1:
After two rounds of revision, the study by Pokrovsky et al has been edited for clarity of presentation and can be accepted for publication after fully addressing the two remaining points.
- Intermittent use of absent/permafrost-free and permafrost-bearing/ permafrost-affected. If you refer to different area definition – clarify. If the term couples refer to the same area – please chose one and use it exclusively through your whole text, figures, tables and supplementary materials. Your text is already heavy on its content, so why complicate its reading even further?
We agree and decided to use the terms “permafrost-affected” and “permafrost-free” rivers throughout the manuscript, and we revised the text, figures, tables and Supplement.
- Proper reference to your “most important finding" presented in Figure 1 is still missing. In your response you wrote: "The seasonal and spatial pattern of DOC export (yield) is certainly the most important finding of this work, and we thought it will be useful to illustrate it in the map." Why don't you refer reader to it from your results section?
We thoroughly describe the DOC results in the relevant section as indicated below:
Contribution of spring, summer and autumn (2 months each) to overall export of DOC during 6-months open-water period (May to October) in 2016 demonstrated the dominant role of spring for both permafrost-free and permafrost-affected rivers (Fig. 2).
and
The DOC yield was minimal in rivers south to 59°N and exhibited a maximum at 61–65°N and 63–65°N in 2015 and 2016, respectively, with overall magnitude of variation by a factor of 4 (Fig. S4 A and 3A, respectively).
Therefore, given that the DOC pattern is adequately described in the results and explicitly illustrated in Figs 2, 3A and S4, we decided not to complicate the presentation and removed the DOC insert from Fig. 1 (which is redundant to more detailed presentation in other figures).
Minor suggestions:
L 120: Building upon? Agree and corrected
L 156: Immediately after filtration, samples for DOC, DIC, major and trace elements were refrigerated and stored at +4C for 1-2 months prior to the analyses. Fixed.
Fig. 1 legend: Google maps (not map) - Corrected.
4.1 Role of… in = Effect of … on..? - Agree and corrected
We thank the reviewer for his/her very insightful review.